# Effect of Grape Pomace Inclusion in the Diet of Ewes Naturally Infected with Gastrointestinal Nematodes During Lactation

**DOI:** 10.3390/pathogens14060560

**Published:** 2025-06-04

**Authors:** Mateus Oliveira Mena, Gustavo Gabriel de Oliveira Trevise, César Cristiano Bassetto, Willinton Hernan Pinchao Pinchao, Helder Louvandini, Ricardo Velludo Gomes de Soutello, Ana Cláudia Alexandre Albuquerque, Alessandro Francisco Talamini do Amarante

**Affiliations:** 1Faculty of Veterinary Medicine and Animal Science, São Paulo State University (UNESP), Botucatu 18618-681, SP, Brazil; gustavo.trevise@unesp.br; 2Department of Biodiversity and Biostatistics, Institute of Biosciences, São Paulo State University (UNESP), Botucatu 18618-689, SP, Brazil; cesar.bassetto@unesp.br (C.C.B.); alessandro.amarante@unesp.br (A.F.T.d.A.); 3Center for Nuclear Energy in Agriculture, Animal Nutrition Laboratory, University of São Paulo (USP), Piracicaba 13416-000, SP, Brazil; wilintonpinchao70@usp.br (W.H.P.P.); louvandini@cena.usp.br (H.L.); 4Faculty of Agricultural and Technological Sciences, São Paulo State University (UNESP), Dracena 17915-899, SP, Brazil; ricardo.vg.soutello@unesp.br; 5Department of Pathology, Reproduction, and One Health, São Paulo State University (UNESP), Jaboticabal 14884-900, SP, Brazil; claudia.albuquerque@unesp.br

**Keywords:** lactating ewes, nutraceuticals, sheep, animal production, agro-food residues

## Abstract

This study evaluated the feasibility of including grape pomace in the diet of Santa Inês ewes during lactation, aiming to reduce the effects of gastrointestinal nematode (GIN) infections and improve productive performance. Grape pomace, which contains phenolic compounds and tannins, was provided as a dietary supplement, replacing 20% of the concentrate over 28 days of lactation, starting on day 7 after lambing. A total of 18 ewes were used, divided into two groups: supplemented (n = 8) and control (n = 10). The supplemented group showed significant reduction in fecal egg count on day 14 of lactation (*p* < 0.05) in comparison with the control group. Lambs from the supplemented group had higher weaning weights (*p* < 0.05). Additionally, the supplemented group showed higher values for packed cell volume and circulating eosinophils, indicating greater resilience to infections. Analysis of anti-L3 IgG against *Haemonchus contortus* revealed no significant differences between the groups. It is concluded that grape pomace supplementation demonstrated potential to improve hematological and productive parameters in lactating ewes, with more evident effects during early lactation. Although the results suggest a possible complementary role in controlling gastrointestinal nematodes, future studies are needed to confirm and optimize this nutritional strategy.

## 1. Introduction

Small ruminant farming is severely affected by numerous factors, among which infections caused by gastrointestinal nematodes (GINs) stand out, leading to significant economic losses due to increased mortality rates and reduced herd productivity [1,2,3].

There are specific phases during which sheep are more susceptible to GIN infections. The final third of gestation and lactation are considered critical periods as nutrient allocation priorities shift toward fetal development and milk production, often at the expense of the immune system [4,5,6,7,8,9,10].

In tropical regions, *Haemonchus contortus*, a hematophagous nematode of the abomasum, is considered the most relevant species due to its high prevalence and pathogenicity [11]. This highly prolific species has females capable of producing between 5000 and 15,000 eggs per day, which are released into the environment through the host’s feces resulting in significant pasture contamination with infective larvae [12]. Another important species is *Trichostrongylus colubriformis*, a parasite of the small intestine that forms tunnels in the intestinal villi to feed on the contents of necrotic epithelial cells. The damage and inflammation caused by this parasite result in villous atrophy, epithelial erosion, and subsequent impairment of nutrient digestion and absorption [13]. In intense infections, they can cause severe enteritis [14]. Similar pathogenic effects are observed in *Cooperia* and *Strongyloides* infections [15]. Infections by *Oesophagostomum* spp. can cause macroscopic lesions with large fibrotic nodules in the intestine, potentially compromising the health and productivity of sheep [16].

The handling of GIN infections is predominantly accomplished through the use of anthelmintics, including albendazole, closantel, levamisole, macrocyclic lactones, and monepantel. Nevertheless, the indiscriminate application of anthelmintics has led to a rise in resistance over recent decades, resulting in a growing prevalence of herds harboring multi-drug-resistant worms [17]. This has motivated the search for alternative control methods that reduce the reliance on anthelmintics, particularly approaches that stimulate the immune response and enhance animal resilience, such as nutritional strategies [18,19,20].

The use of food byproducts such as grape pomace, a residue generated by the wine industry, has gained attention as an alternative ingredient. Its application is promising not only for reducing feeding costs but also for promoting sustainability by repurposing materials that would otherwise be discarded [21,22]. The inclusion of desiccated and milled grape pomace into animal feed and rations offers a feasible method to reduce the environmental effect and enhance the value of this residue [23].

Grape pomace comprises seeds, leftover pulp, and stems, which are abundant in fiber (49.37%) and phenolic chemicals (35.35 mg of gallic acid per gramme), including tannins, carotenoids, and other bioactive antioxidants [23,24]. The dried pomace has a significant concentration of phenolics, measuring 3.95 mg/g (tannic acid equivalent), and tannins, totaling 1.92 mg/g (tannic acid equivalent), per 100 mg. The hydroalcoholic extract of grape pomace has in vitro anthelmintic activity, significantly reducing the hatching, development, and motility of free-living stages of *H. contortus* [25].

Santa Inês lambs artificially infected with *H. contortus* and supplemented with grape pomace, replacing 20% of the concentrate provided to the animals, showed significantly lower fecal egg counts (FECs) compared to the group that did not receive this supplement. It was also observed that these animals harbored *Haemonchus* females with shorter lengths and fewer eggs in the uterus [26]. Furthermore, grape pomace contained 15.57% crude protein, allowing supplemented animals to achieve weight gains similar to those receiving concentrate without the supplement, highlighting the potential of this byproduct as a dietary supplement (ibidem).

The benefits of including grape pomace in the diet of lambs [26] motivated this study, which aimed to evaluate the feasibility of including this product in the diet of Santa Inês ewes during lactation, with a focus on the nutraceutical benefits of the product, particularly in enhancing resistance/resilience against gastrointestinal nematode infections.

## 2. Materials and Methods

### 2.1. Location, Animals, and Experimental Design

All procedures involving animals in this study comply with international ethical standards and were approved by the Animal Use Ethics Committee of the Institute of Biosciences at São Paulo State University (IBB/UNESP), under protocol CEUA n° 7450220324 (ID 000729).

The experiment was carried out in the Veterinary Helminthology Laboratory’s experimental site, IBB (Institute of Biosciences), UNESP (São Paulo State University), situated in Botucatu, SP, Brazil (22°53′16″ S and 48°29′57″ O, altitude 884 m), from July 2023 to April 2024.

In July 2023, 2 rams were placed in a paddock with 27 multiparous Santa Inês ewes, with an average age of three years. All ewes had pre-existing natural infections with gastrointestinal nematodes (GINs), confirmed by fecal egg counts (FECs) prior to the experiment. In September 2023, after natural mating, pregnancy was confirmed in 25 ewes by ultrasound, of which 20 were selected for the experiment. Throughout the experiment, the sheep were kept in a 3840 m^2^ area paddock containing *Urochloa decumbens*, which maintained natural larval contamination and ensured continuous GIN exposure. The sheep were provided with water troughs, mineral salt feeders (ad libitum), and concentrate feed (70% ground corn and 30% soybean meal), supplied daily in collective troughs at a rate equivalent to 2% of the ewes’ body weight. The selected ewes lambed over 29 days, between 7 December 2023 and 5 January 2024.

As the ewes lambed, they were evenly distributed into two groups based on the birth of single or twin lambs, in a completely randomized design, with 10 ewes in each group.

However, in the supplemented group, data from two ewes and their respective lambs were excluded: one died 55 days after lambing due to an undetermined cause and the other developed mastitis. Therefore, the supplemented group consisted of eight ewes.

Seven days after lambing (D7), each ewe was placed in an individual pen with water ad libitum where they received daily concentrate feed, as outlined in Table 1. Immediately after confirming the complete consumption of the concentrate, the animals from both groups were released into the same paddock where they grazed together. This procedure was carried out over 28 days, starting on day 7 after lambing and ending on day 35 after lambing (D7 to D35).

After this period, the animals continued to be fed in collective troughs with the same diet as at the beginning of the experiment. The lambs accessed the feeders with their mothers and started ingesting little quantities of feed as weaning neared.

When the lambs reached 60 days of age, they were weaned. The ewes were moved to a separate paddock away from the lambs, where they did not receive concentrate feed to facilitate milk drying.

### 2.2. Grape Pomace Preparation

Syrah grapes destined for wine production were harvested in August 2023. The grape pomace was collected from a winery located in São Roque, São Paulo, Brazil (23°35′41.2″ S 47°09′41.6″ W, altitude 880 m). The material was dried in a forced-air oven at 50 °C for approximately 24 h, then ground and stored at −20 °C until use.

### 2.3. Measurement of Total Phenolics and Tannins in Grape Pomace

Total phenolic (TP) concentrations were determined using the Folin–Ciocalteu reagent method as described by Makkar [27]. Total tannins (TTs) were measured as the difference between TP concentrations before and after treatment with insoluble polyvinylpolypyrrolidone, following Makkar’s [28] protocol with tannic acid as standard. Condensed tannins (CTs) were quantified using the butanol–HCl method [27], with leucocyanidin as reference standard.

### 2.4. Fecal Sample Collection and Coprological Exams

Fecal samples were collected directly from the rectal ampulla of each animal, placed in pre-labeled polyethylene bags, and refrigerated (4 °C) until processing. Collections followed Table 2’s schedule, beginning at least 38 days before each lamb’s birth and concluding 28 days post-weaning, for FECs and coprocultures. FECs and coprocultures were performed according to Ueno and Gonçalves [29], including the identification of infective larvae. As lamb births occurred on different dates, weekly coprocultures (processed every Friday) contained refrigerated (4 °C) samples collected from Saturday to Friday, with two separate cultures processed per treatment group.

### 2.5. Hematological Analyses

Blood samples were collected per Table 2’s schedule via jugular venipuncture into EDTA vacutainers. Packed cell volume (PCV) was determined by microhematocrit centrifugation. Total plasma protein (TPP) was measured by refractometry (SPR-Atago). Eosinophils were quantified using a Neubauer chamber after Carpentier’s solution staining [30], expressed as cells/μL blood. Samples were centrifuged at 1000× *g* for 15 min to obtain plasma, which was stored at −20 °C for subsequent analyses.

### 2.6. Anti-H. contortus L3 Immunoglobulin G (IgG)

To determine IgG levels against L3-soluble extract of *H. contortus*. The L3 extract (antigen) production followed earlier instructions [31]. Following a method previously established to measure the parasite-specific plasma IgG levels [32], with some modifications: In short, the plates were coated with 2 μg of antigen/mL, washed three times, rotated 180 degrees, and then washed three times more. Following the addition of diluted plasma and subsequent washing, the plates were incubated with a 1:20,000 diluted peroxidase-conjugated rabbit anti-sheep IgG, washed, and incubated with OPD (o-Phenylenediamine, Ref: P5412, Sigma) after 15 min the plate was read at 405 nm. The negative control (NC) sample used was from a worm-free animal, previously described [33], and the plasma positive control (PC) sample used was from an ewe naturally infected. Results were expressed as the percentage of the PC plasma sample's optical density (OD) value [34].

### 2.7. Bromatological Analyses

Seven pasture samples were collected during paddock rotations (sampling the new paddock on transfer days). Concentrate and grape pomace samples were collected and frozen until analysis. Samples underwent bromatological analyses for dry matter (DM), organic matter (OM), crude protein (CP), ether extract (EE), and mineral matter (MM) following AOAC [35] protocols. Neutral detergent fiber (NDF) and acid detergent fiber (ADF) contents were determined via sequential washing using a fiber analyzer (TE-149 Tecnal, Piracicaba-SP) and F57 filter bags (Ankom Technology Corp., Macedon, NY, USA), following Van Soest, Robertson, and Lewis [36] as modified by Mertens [37].

### 2.8. Weight Gain

Ewes were weighed per Table 2 schedule each morning before concentrate feeding. Lambs were weighed at birth and weaning.

### 2.9. Meteorological Data

Daily mean temperature and rainfall were recorded. Rainfall data came from an on-site pluviometer, while temperature data were obtained from UNESP’s Lageado Experimental Farm weather station (7.7 km from the experimental site).

### 2.10. Statistical Analysis

Data were analyzed using Statistical Analysis System version 9.4 (SAS Institute, Cary, NC, USA). Repeated measures were evaluated with PROC MIXED. When the data did not present a normal distribution according to the normality test, they were transformed by log10(x + 1). Means were compared by *t*-test (α = 0.05). The results section presents arithmetic means (±standard error). Only significant time × treatment interactions are reported.

## 3. Results

### 3.1. Dietary Nutrients

A total of 227 kg of grape pomace was obtained, which was reduced to 101 kg after drying, representing a 44% reduction. Regarding consumption in the grape pomace-supplemented group, from the first day (D7) until the last day of supplementation (D35) all ewes in the supplemented group consumed the provided concentrate. Ewes of the control group also consumed all the concentrate.

Averages of the seven pasture samples collected from the paddocks grazed by sheep, together with the bromatological composition of the grape pomace and concentrate and the measurement of total phenolics, total tannins, and condensed tannins, are shown in Table 3.

### 3.2. Fecal Examination

Fecal egg count (FEC) means varied between the supplemented and control groups throughout the experiment (Table 4). At study initiation (D-38, prepartum), both groups showed low FEC means. The supplemented group averaged 75 (±36.60) EPG, whilst controls had 260 (±195.62) EPG, with no significant difference (*p* = 0.7507). Subsequently, values increased and at day 7 (D7) during lactation the supplemented group mean peaked reaching 1663 (±551.60) EPG, while the control averaged 1790 (±381.65) EPG (*p* = 0.7943).

By the next collection (D14), the supplemented group showed a marked FEC reduction (975 ± 413.5 EPG), while the control increased to 2960 (±1192.87) EPG, demonstrating significant differences between groups (*p* = 0.0087). This latter value represented the highest mean observed in the control.

From D7 onward, the supplemented group means remained lower than the control until study completion, except at D77 when supplemented ewes averaged 550 (±281.58) EPG versus 440 (±166.13) EPG in the control (*p* = 0.8616).

### 3.3. Coproculture

The coproculture analysis revealed a predominance of *Haemonchus* spp. in both groups (supplemented and control), followed by *Oesophagostomum* spp., *Trichostrongylus* spp., and *Cooperia* spp. Regarding monthly distribution, *Haemonchus* spp. predominated in coprocultures during the initial months of the experiment, with a mean percentage of 90%. However, by the study’s conclusion there was a significant increase in the proportion of *Oesophagostomum* spp., reaching 100% in one control group sampling. *Trichostrongylus* spp. and *Cooperia* spp. appeared in intermediate samplings with mean percentages of 11% and 13%, respectively.

Mean ewe body weights are presented in Figure 1. No significant differences (*p* > 0.05) were observed between the supplemented and control groups at any evaluation period. Post-weaning weight reduction occurred in all ewes, attributable to the discontinuation of concentrate supplementation to facilitate milk drying and prevent mastitis incidence.

### 3.4. Packed Cell Volume and Total Plasma Protein

Significant differences (*p* < 0.05) between groups were only observed in PCV values on D14 when the supplemented group showed a significantly higher PCV (*p* = 0.0399). On this day, mean PCV values were 32% (±0.98) for the supplemented group and 28.9% (±0.98) for the control group (Figure 2).

The results regarding TPP values were similar, showing no significant differences between the groups over time (Figure 3).

### 3.5. Blood Eosinophil Count and Anti-H. contortus L3 IgG Analysis

Blood eosinophil counts showed no significant differences (*p* > 0.05) between groups. However, the supplemented group generally exhibited higher mean values compared to controls, except at D35 when the supplemented group averaged 2319 (±498.91) eosinophils/μL versus 2418 (±582.87) eosinophils/μL in controls (Figure 4).

Anti-*H. contortus* L3 IgG analysis revealed fluctuating antibody levels between supplemented and control groups throughout the study, though without significant differences (*p* > 0.05) (Figure 5).

### 3.6. Body Weight of Lambs

Single-born lambs showed a higher birth weight compared to twins (*p* = 0.0193), with no significant difference between the experimental groups (*p* = 0.2675). At weaning, weight was higher in lambs from supplemented ewes, with a significant effect of birth type (*p* = 0.0008) and significant interaction between birth type and group (*p* < 0.0001). Weight gain was higher in lambs from supplemented ewes. There was a significant effect of birth type (*p* = 0.0029) and a tendency toward significance for supplementation (*p* = 0.0539) (Table 5).

### 3.7. Meteorological Data

The recording of meteorological data began seven days after the first lambing (D7), which was on 14 December 2023, and continued until 5 April 2024, the day of the last collection, coinciding with the summer and autumn seasons. The daily average rainfall was 2.71 mm, and the average temperature was 24.03 °C. Of the 17 days with observations in December, a total of 19.38 mm of rain was recorded, with a daily average temperature of 25.57 °C. This was the month with the highest temperatures. The month with the highest rainfall was January, with a total of 182.50 mm of rain and an average daily temperature of 23.46 °C (Figure 6).

## 4. Discussion

The results of this study indicate that although grape pomace supplementation did not provide significant protection against GINs it showed potential to reduce FECs during the periparturient relaxation of immunity without compromising the productive performance of the ewes. In vitro studies by Soares et al. [25] demonstrated the anthelmintic activity of grape pomace extract against *H. contortus* eggs and larvae, likely due to the presence of phenolic compounds. In 100 g of dried pomace, the total phenolic content and total tannins in dry matter were 3.95 mg/g and 1.92 mg/g (tannic acid equivalent), respectively.

The relaxation of the immune response due to the peripartum phenomenon was evident in both groups, which showed elevated FECs between D17 and D14. The supplemented and control groups reached their highest mean FEC on D7 and D14, respectively, followed by a progressive decline throughout the experiment. Even ewes supplemented with high levels of metabolizable protein exhibit periparturient relaxation of immunity, leading to increased *H. contortus* egg shedding [9]. López-Leyva et al. [38] reported that lactating ewes, regardless of diet, showed a significant increase in FEC (2709 EPG). However, those fed high-protein diets (15% crude protein) had reduced FEC values compared to those receiving low amounts of protein (8% CP). Beasley et al. [39] linked periparturient relaxation of immunity to suppressed eosinophils and local antibodies in Merino ewes. Our results reinforce that lactation is the primary driver of periparturient relaxation of immunity as grape pomace supplementation (rich in tannins) mitigated but did not eliminate the FEC rise. This highlights the need for combined strategies (nutrition, management, and genetics) for effective periparturient relaxation of immunity control.

Tannins have direct anthelmintic activity by inhibiting egg hatching, larval motility, and adult nematode fertility, consequently reducing FECs [25,40]. The tannins in grape pomace may also enhance ruminant immune responses. They can bind to dietary proteins, forming complexes that protect them from ruminal degradation, increasing amino acid availability in the small intestine and improving nutrient absorption [41]. These indirect effects may have contributed to the greater resilience of supplemented ewes, reflected in lower FECs and higher PCVs and eosinophil counts in the supplemented group. Similar results were observed in lambs artificially infected with *H. contortus* and supplemented with grape pomace for 28 days, which had significantly lower FECs than the control group [26], reinforcing the benefits of grape pomace in sheep nutrition.

It is important to note that this study spanned spring, summer, and autumn—seasons where frequent rainfall and warm temperatures favor the development and survival of L3 larvae [42]. The predominance of *Haemonchus* spp. in coprocultures aligns with previous observations by Wilmsen et al. [43], who reported its high prevalence in sheep from the same region. Despite its high pathogenicity and potential to cause anemia, no clinical cases of haemonchosis were observed, even in animals with a high FEC (e.g., one control ewe with 12,300 EPG and 25% PCV on D14). This suggests that the diet enhanced resilience.

Regarding hematological parameters, while most values remained within normal ranges without significant differences between groups, the supplemented group showed a significantly higher PCV on D14. A similar outcome was reported in a study using 10% fresh Arabica coffee byproduct feed, which also did not significantly reduce FECs in infected lambs [44] despite containing 13.95 g/100 g total phenolics and 2.28 g/100 g condensed tannins [45]. This suggests tannin efficacy may vary by source and concentration.

While grape pomace supplementation did not affect ewes’ productive performance, lambs from supplemented ewes had higher mean weaning weights, possibly due to improved milk production. This aligns with findings by Dove et al. [46], where lambs from beet-pulp-supplemented ewes gained 40–60 g/day more than the controls. Undernutrition or overnutrition during gestation can influence lamb muscle development and lifelong growth capacity [47,48].

In this study, grape pomace did not significantly enhance immunological variables (e.g., eosinophil counts, IgG levels), likely due to the short supplementation period. However, the absence of negative effects supports its use as a partial feed replacement without compromising health or performance.

The use of byproducts like grape pomace is a promising alternative for ruminant nutrition, particularly amid rising anthelmintic resistance. Beyond reducing feed costs by up to 33.8% [49], it promotes sustainable livestock production by repurposing waste [50]. Further research should investigate mechanisms of action, optimal dosages, and supplementation timing, especially during critical phases like lactation.

## 5. Conclusions

Grape pomace supplementation demonstrated potential to improve hematological and productive parameters in lactating ewes, with more evident effects during early lactation. Although the results suggest a possible complementary role in controlling gastrointestinal nematodes, future studies are needed to confirm and optimize this nutritional strategy.

## Figures and Tables

**Figure 1 pathogens-14-00560-f001:**
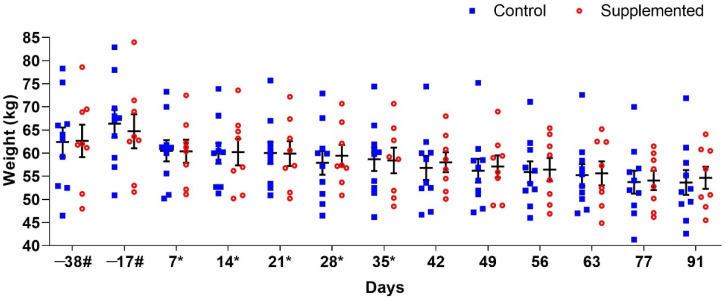
Mean body weight (±standard error) of ewes supplemented with grape pomace versus the control group during the experimental period. # Approximate days. * Supply of grape pomace for the supplemented group.

**Figure 2 pathogens-14-00560-f002:**
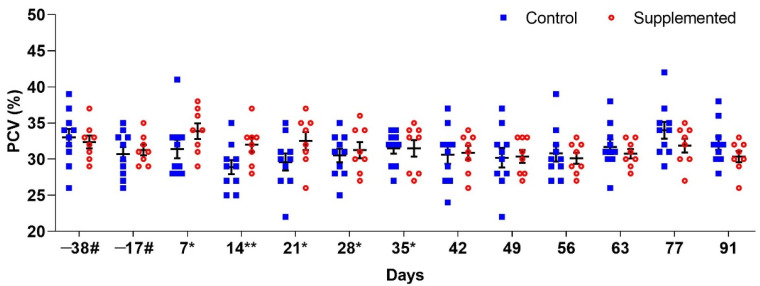
Mean packed cell volume (PCV) (±standard error) in ewes supplemented with grape pomace versus the control group. # Approximate days. * Supply of grape pomace for the supplemented group. ** Statistically significant difference.

**Figure 3 pathogens-14-00560-f003:**
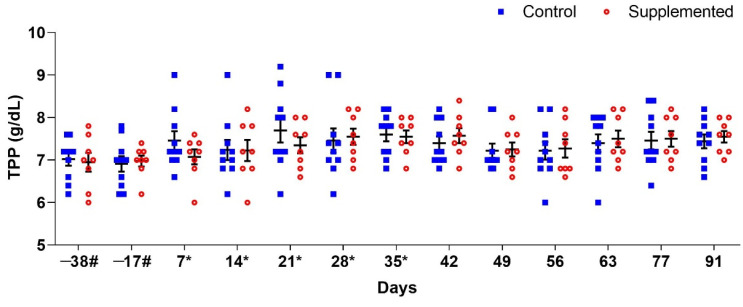
Mean total plasma protein (TPP) (±standard error) concentrations (g/dL) in ewes supplemented with grape pomace versus the control group. # Approximate days. * Supply of grape pomace for the supplemented group.

**Figure 4 pathogens-14-00560-f004:**
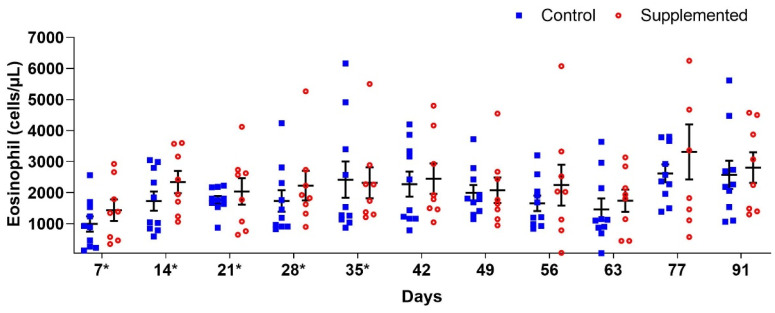
Mean eosinophil counts per µL of blood (±standard error) in ewes supplemented with grape pomace versus the control group. * Supply of grape pomace for the supplemented group.

**Figure 5 pathogens-14-00560-f005:**
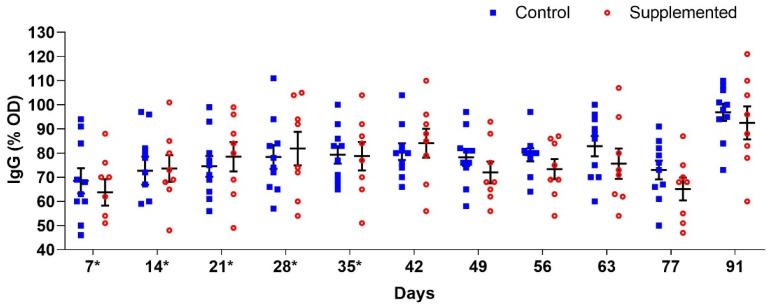
Mean anti-*H. contortus* L3 IgG levels (percentage of optical density, OD) (±standard error) in the serum of ewes supplemented with grape pomace versus the control group. * Supply of grape pomace for the supplemented group.

**Figure 6 pathogens-14-00560-f006:**
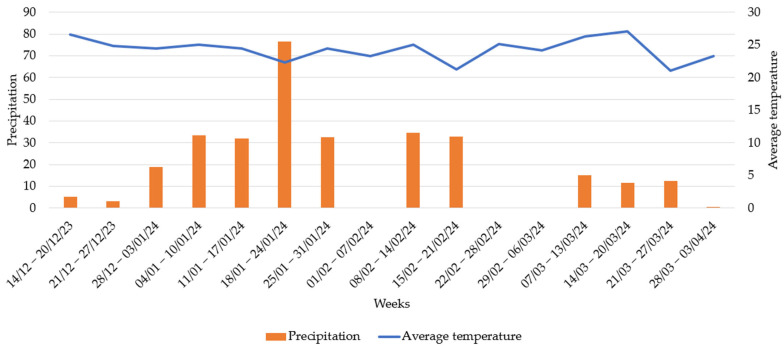
Meteorological data of weekly mean temperature (°C) and weekly accumulated precipitation (mm).

**Table 1 pathogens-14-00560-t001:** Amount in kilograms (kg) of concentrate and grape pomace provided individually to ewes from day 7 postpartum to day 35.

Groups	Type of Birth	Concentrate (kg)	Grape Pomace (kg)
Supplemented (n = 8)	Single (n = 5)	0.8	0.2
	Twin (n = 3)	1.0	0.2
Control (n = 10)	Single (n = 6)	1.0	0
	Twin (n = 4)	1.2	0

Concentrate composed of 70% ground corn and 30% soybean meal.

**Table 2 pathogens-14-00560-t002:** Experimental timeline.

Procedures	Days													
	−38 *	−17 *	0	7	14	21	28	35	42	49	56	63	77	91
Prepartum sampling	X	X												
Parturition day			X											
Lactation sampling **				X	X	X	X	X	X	X	X	X		
Post-weaning sampling													X	X
Grape pomace feeding				X	X	X	X	X						

* Approximate days. ** Blood and fecal sampling plus body weight measurements.

**Table 3 pathogens-14-00560-t003:** Nutritional composition of grape pomace, pasture, and concentrate provided to ewes supplemented with grape pomace or control diet and measurement of total phenolics and tannins in grape pomace.

Variables (%)	Grape Pomace	Concentrate	Pasture
DM	93.59	91.16	86.02
CP	12.61	18.71	10.13
NDF	45.42	17.17	72.03
ADF	66.50	12.06	46.85
EE	5.71	3.19	2.55
MM	10.89	3.38	8.59
Total phenolic compounds ^a^	47.87	-	-
Total tannins ^a^	42.21	-	-
Condensed tannins ^b^	2.63	-	-

Legend: DM—dry matter; MM—mineral matter; CP—crude protein; EE—ether extract; NDF—neutral detergent fiber; ADF—acid detergent fiber. ^a^—Values expressed in equivalent gram of tannic acid/kg dry matter. ^b^—Values expressed in equivalent gram of leucocyanidin/kg dry matter.

**Table 4 pathogens-14-00560-t004:** Mean (±standard error) fecal egg counts (FECs) in ewes supplemented with grape pomace versus controls.

Period	Day	Supplemented (n = 8)	Control (n = 10)	*p*-Value
Prepartum	−38 #	75.0 ± 36.60	260.0 ± 195.62	0.7507
	−17 #	1625.0 ± 649.38	520.0 ± 174.99	0.0864
Lactation (Weekly sampling)	7 *	1662.5 ± 551.60	1790.0 ± 381.65	0.7943
	14 *	975.0 ± 413.50	2960.0 ± 1192.87	0.0087
	21 *	725.0 ± 372.13	1970.0 ± 921.96	0.0856
	28 *	762.5 ± 402.64	1370.0 ± 805.13	0.4305
	35 *	475.0 ± 256.87	1200.0 ± 516.61	0.2520
	42	437.5 ± 155.77	1300.0 ± 587.08	0.1894
	49	237.5 ± 98.08	670.0 ± 255.63	0.4570
	56	337.5 ± 129.47	500.0 ± 197.77	0.7816
	63	250.0 ± 96.36	320.0 ± 121.84	0.9019
Post-Weaning (Biweekly sampling)	77	550.0 ± 281.58	440.0 ± 166.13	0.8616
	91	437.5 ± 186.07	540.0 ± 260.00	0.8749

Legend: # Approximate days; * supply of grape pomace for the supplemented group.

**Table 5 pathogens-14-00560-t005:** Body weight of lambs (±standard error) born to ewes supplemented with grape pomace or in the control group.

Weight	Supplemented (n = 11)		Control (n = 12)		*p*-Value		
	Single (5)	Twin (6)	Single (7)	Twin (5)	Type of Birth	Group	Type of Birth × Group
Birth	4.2 ± 0.42	3.4 ± 0.24	4.6 ± 0.33	3.8 ± 0.24	0.0193	0.2675	0.7602
Weaning	19.5 ± 1.29	15.2 ± 1.65	18.2 ± 0.73	11.8 ± 1.62	0.0008	0.0942	<0.0001
Weight gain	15.2 ± 0.94	11.8 ± 1.60	13.6 ± 0.77	8.0 ± 1.83	0.0029	0.0539	0.4394

## Data Availability

The data presented in this study are available on request from the corresponding author.

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
