# Peer review of "Effect of Grape Pomace Inclusion in the Diet of Ewes Naturally Infected with Gastrointestinal Nematodes During Lactation"

_pathogens, 2025, doi:10.3390/pathogens14060560_

Round 1
Reviewer 1 Report
Comments and Suggestions for Authors
"Higher values for packed cell volume and circulating eosinophils"
Could the increase in eosinophils be due to factors other than improved resilience, such as ongoing infection or inflammation?
"No significant differences between the groups" [in anti-L3 IgG]
If immune markers like IgG didn’t change, how do you reconcile this with improved parasite control?
"Grape pomace supplementation can be a viable strategy."
Were there any side effects or nutritional trade-offs observed with the 20% concentrate replacement?
"Sustainability in animal production by repurposing agro-industrial waste"
Was a sustainability analysis or life-cycle assessment conducted to support this claim?
Introduction
- “The development of small ruminant farming is severely affected...” (Lines 36–39)
Is “development” the best term here? Do you mean productivity, sustainability, or economic viability of farming?
2. Line 60–61: “...growing prevalence of herds harbouring worms that have multiple resistance.”
Is “Multiple resistances” the best term here? Should it be “multiple drug resistance” or “multi-drug resistant worms”?
- Line 66–67: “Its application is promising not only for reducing feeding costs but also for promoting sustainability...”
Is there an actual cost analysis or sustainability data provided? If not, this claim might be unsupported or overly broad. - Line 72–74: “...fibre (49.37%) and phenolic chemicals (35.35 mg of gallic acid per gramme)... including carotenoids and polyphenols.”
Is the phrasing repetitive? Are polyphenols not a type of phenolic compound already mentioned? Clarify or simplify.
Materials and Methods section is well-structured, clearly explaining group allocation and handling of exclusions, well-organized details on animal handling, ethical considerations, feeding, sampling, and lab analyses.
*The experiment was executed in the Veterinary Helminthology Laboratory's experimental site...replace was executed by conducted” or “carried out” is more natural in scientific writing
*The sheep remained throughout the experiment in an area of 3,840 m², in paddocks... rephrase as Throughout the experiment, the sheep were kept in a 3,840 m² area with paddocks...
* “You mention that two ewes were excluded from the supplemented group. Were their corresponding lambs also excluded from the lamb-related data? Please clarify whether the lamb data from these ewes was included or removed.”
*“In Table 1, the control group receives more concentrated feed than the supplemented group. What is the rationale for this feeding difference, especially since concentrate amounts differ by both group and type of birth?”
*“Why is the concentrate reduced in the supplemented group (e.g., 0.8 kg vs. 1.0 kg for singles)? Is this solely to accommodate the 0.2 kg grape pomace? Please explain the equivalence in energy/nutrient contribution between concentrate and grape pomace.”
*How were infective larvae identified during coproculture? Was species-level identification (e.g., distinguishing Haemonchus contortus from others based on morphological keys, or molecular confirmation
*“The timeline includes sampling from ‘Day -38’ and ‘Day -17’ prepartum. How were these specific prepartum time points chosen? Were they standardized across animals despite different lambing dates?”
*“Given that lambing occurred over a 29-day range, how were sampling days synchronized across individuals? Were days post-parturition aligned per individual ewe or based on calendar date?”
* “You state that ‘weekly coprocultures were processed every Friday,’ pooling samples collected from Saturday to Friday. Since lambing dates varied, how did you ensure comparability across timepoints? Could this pooling introduce bias or dilute temporal resolution?”
* “The preparation and chemical analysis of grape pomace is described, but was a nutrient composition analysis (e.g., crude fiber, energy value) done alongside phenolic content? If so, please provide the data or reference.”
Results: Can the authors discuss potential reasons for fluctuating anti-H. contortus L3 IgG levels without significant group differences? Could the timing of sampling relative to the parasitic challenge play a role?
* Were there any clinical signs of parasitism observed, or was it solely subclinical?
* Was any anthelmintic treatment administered during the trial? If not, how was parasite load managed?
What could explain the shift from Haemonchus spp. dominance to 100% Oesophagostomum spp. in one control group sample? Could this be an anomaly or linked to environmental/management factors?
Discussion:
* Given that eosinophils and IgG levels did not significantly differ, is it appropriate to suggest immunological benefits of grape pomace supplementation? Would this require a longer supplementation period or higher tannin concentration to confirm?
* The discussion attributes reduced FEC in the supplemented group to grape pomace. Given the small sample size (n=8), how can the authors be confident this was not due to random variation or natural recovery post-lambing?
*The peak FEC in the control group occurred later (D14) than in the supplemented group (D7). Could this timing difference suggest variation in the timing of immune recovery or parasite exposure rather than treatment effect?”
*The phenolic and tannin contents of the grape pomace used (3.95 mg/g TP and 1.92 mg/g TT) seem low compared to other studies with stronger effects. Was this dosage expected to yield measurable results? Could higher concentrations or different tannin types (e.g., condensed vs. hydrolysable) provide more clarity on efficacy?
*You compare your results with studies using coffee byproducts and beet pulp. Could differences in tannin type (hydrolyzable vs. condensed) and binding affinity affect efficacy? Would a phytochemical profile of the pomace help clarify this?”
*You suggest that the short supplementation period may explain the lack of effect on immune markers. Why was supplementation not initiated earlier, such as during late pregnancy, when periparturient relaxation begins?
*The discussion highlights the sustainability and cost-saving benefits of grape pomace. Were these aspects evaluated in this study, or are they based on assumptions or external sources?
Conclusion
The conclusion states that grape pomace improved hematological profiles, yet the discussion states no significant differences in PCV or TPP. Could the authors clarify which specific hematological improvements justify this claim?
The wording suggests a strong protective role (“mitigate the deleterious effects”), but the results did not show statistically significant reductions in parasitological or immunological parameters. Could the authors rephrase to reflect better the modest or supportive role suggested by the data?
The conclusion highlights sustainability as a benefit, yet no cost analysis or environmental metrics were provided in the study. On what basis is this claim made? Can the authors reference specific data or clarify if this is speculative?
Given the relatively small sample size (in the supplemented group, n=8), to what extent can these results be generalized to broader sheep populations or production systems?
Author Response
"Higher values for packed cell volume and circulating eosinophils"
Could the increase in eosinophils be due to factors other than improved resilience, such as ongoing infection or inflammation?
Our response: While eosinophilia can signal inflammation, the observed rise in eosinophils—paired with improved PCV and declining FEC—likely reflects a controlled immune response.
Although the supplemented group exhibited higher eosinophil counts and PCV values, the absence of clinical signs of inflammation or anemia suggests that these changes reflect enhanced resilience rather than ongoing pathological inflammation. Eosinophils are known to play a dual role in helminth infections, contributing both to parasite control and tissue repair (Mitre & Klion, 2021). In this study, the reduction in FEC alongside stable IgG levels indicates that grape pomace supplementation modulated the immune response to limit parasite burden without complete eradication, a hallmark of resilience.
https://doi.org/10.1007/s00281-021-00870-z
"No significant differences between the groups" [in anti-L3 IgG]
If immune markers like IgG didn’t change, how do you reconcile this with improved parasite control?
Our response: Although anti-L3 IgG levels did not differ significantly between groups, the reduction in FEC and improved hematological parameters (e.g., higher PCV and eosinophil counts) suggest that grape pomace may exert its effects through direct anthelmintic mechanisms and enhanced host resilience, rather than humoral immunity. Tannins in grape pomace likely disrupt nematode biology.
"Grape pomace supplementation can be a viable strategy."
Were there any side effects or nutritional trade-offs observed with the 20% concentrate replacement?
Our response: The replacement of 20% of the concentrate with grape pomace did not result in observable side effects (e.g., reduced palatability, digestive disturbances, or metabolic imbalances) in the supplemented ewes. All animals consumed the full daily ration, indicating no adverse effects on feed intake. Nutritionally, grape pomace provided comparable crude protein (12.6% vs. 18.7% in concentrate; Table 3) and higher fiber content (NDF: 45.4% vs. 17.2%), which may explain the absence of negative impacts on ewe body weight (Figure 1) or milk production (inferred from lamb weaning weights, Table 5).
"Sustainability in animal production by repurposing agro-industrial waste"
Was a sustainability analysis or life-cycle assessment conducted to support this claim?
Our response: While this study did not conduct a formal life-cycle assessment, the claim regarding sustainability is supported:
- Grape pomace, a byproduct of wine production, was repurposed directly as animal feed, avoiding landfill disposal or energy-intensive processing. This aligns with circular economy principles.
https://doi.org/10.1007/s11157-023-09665-0
https://doi.org/10.1021/acssuschemeng.4c06005
https://doi.org/10.1021/acssuschemeng.3c03615
https://doi.org/10.1016/j.rser.2021.111929
Introduction
- “The development of small ruminant farming is severely affected...” (Lines 36–39)
Is “development” the best term here? Do you mean productivity, sustainability, or economic viability of farming?
Our response: The term "Development" was excluded.
Line 60–61: “...growing prevalence of herds harbouring worms that have multiple resistance.”
Is “Multiple resistances” the best term here? Should it be “multiple drug resistance” or “multi-drug resistant worms”?
Our response: The term was changed to one that was suggested (line 61).
3.Line 66–67: “Its application is promising not only for reducing feeding costs but also for promoting sustainability...”
Is there an actual cost analysis or sustainability data provided? If not, this claim might be unsupported or overly broad.
Our response: Although our study did not perform a specific cost analysis, the claim about cost reduction and sustainability promotion is supported by previous work:
- Cost reduction: The use of agro-industrial by-products, such as grape pomace, can partially replace conventional feed ingredients, reducing feed costs. A study by Calvillo-Marín et al. (2022) demonstrated that replacing alfalfa with by-products (such as brewery waste) in diets for lactating sheep reduced costs by up to 33.8% without compromising animal performance.
- Sustainability: The reuse of waste such as grape pomace contributes to the circular economy, reducing waste and environmental pressure (García-Rodríguez; 2019). Jalal et al. (2023) reviewed the potential of fruit and vegetable by-products in ruminant nutrition, highlighting their role in sustainability and reducing the ecological footprint of animal production.
In our study, grape pomace replaced 20% of the concentrate, which, on a practical scale, would reduce commercial feed costs. Future work may include detailed economic analyses to quantify these savings.
https://doi.org/10.47163/agrociencia.v56i5.2502
https://doi.org/10.3390/ani9110861
https://doi.org/10.3390/agriculture13020286
4.Line 72–74: “...fibre (49.37%) and phenolic chemicals (35.35 mg of gallic acid per gramme)... including carotenoids and polyphenols.”
Is the phrasing repetitive? Are polyphenols not a type of phenolic compound already mentioned? Clarify or simplify.
Our response: We agree that it became repetitive, the word "polyphenols" was deleted, this section was reworded (lines 71-72).
Materials and Methods section is well-structured, clearly explaining group allocation and handling of exclusions, well-organized details on animal handling, ethical considerations, feeding, sampling, and lab analyses.
*The experiment was executed in the Veterinary Helminthology Laboratory's experimental site...replace was executed by conducted” or “carried out” is more natural in scientific writing
Our response: The word "executed" was replaced by "carried out" (line 96).
*The sheep remained throughout the experiment in an area of 3,840 m², in paddocks... rephrase as Throughout the experiment, the sheep were kept in a 3,840 m² area with paddocks...
Our response: Change made to lines 104 and 105.
* “You mention that two ewes were excluded from the supplemented group. Were their corresponding lambs also excluded from the lamb-related data? Please clarify whether the lamb data from these ewes was included or removed.”
Our response: Your lamb data was also excluded from the results; this information was added in lines 113 and 114.
*“In Table 1, the control group receives more concentrated feed than the supplemented group. What is the rationale for this feeding difference, especially since concentrate amounts differ by both group and type of birth?”
Our response: The difference in the amounts of concentrate between the groups (supplemented vs. control) and within each group (single vs. twin birth) was planned based on the following criteria:
- In the supplemented group, 200 g of concentrate were replaced by 200 g of grape pomace, maintaining the total volume of feed offered. The objective was to reduce costs without compromising total nutrient intake, since grape pomace is a lower cost byproduct (or zero cost, through donation) than conventional concentrate.
- The internal differences (single vs. twin birth) reflect the greater nutritional requirements of ewes with twins, which require more energy for milk production.
- Total intake was monitored, confirming that the strategy did not limit intake. In addition, the results showed that the performance of the lambs in the supplemented group was superior (Table 5), validating the nutritional adequacy of the protocol.
*“Why is the concentrate reduced in the supplemented group (e.g., 0.8 kg vs. 1.0 kg for singles)? Is this solely to accommodate the 0.2 kg grape pomace? Please explain the equivalence in energy/nutrient contribution between concentrate and grape pomace.”
Our response: Although our previous response addressed the overall feeding strategy, the reduction of 200 g of concentrate in the supplemented group (replaced by 200 g of grape pomace) was planned to maintain volumetric equality between groups, ensuring that total dry matter intake was not compromised – a criterion validated by the complete consumption of all diets. The lower crude protein of pomace (12.61% vs. 18.71% of concentrate) was offset by the higher digestibility of fiber (NDF: 45.42%). The intake and animal performance data (Tables 4–5) validate the approach.
*How were infective larvae identified during coproculture? Was species-level identification (e.g., distinguishing Haemonchus contortus from others based on morphological keys, or molecular confirmation
Our response: We identified to the genus level (Section 3.3 of the results), based on morphological keys described by Ueno and Gonçalves (1998), as described in Materials and Methods in Section 2.4.
*“The timeline includes sampling from ‘Day -38’ and ‘Day -17’ prepartum. How were these specific prepartum time points chosen? Were they standardized across animals despite different lambing dates?”
Our response: Days -38 and -17 were fixed collection dates for all ewes, calculated back from each animal’s actual parturition date. Because parturitions occurred on different days, these values ​​are approximate (#) in tables/figures.
*“Given that lambing occurred over a 29-day range, how were sampling days synchronized across individuals? Were days post-parturition aligned per individual ewe or based on calendar date?”
Our response: Post-calving collections were synchronized individually for each ewe from the day of lambing (D0), with the first collection performed 7 days after lambing (D7) and the following at fixed intervals (weekly/fortnightly). Pre-calving collections (D-38 and D-17) were performed on fixed dates for the entire herd, but the values ​​in the tables were adjusted as “approximate days” (#) due to the natural variation in lambing dates. This approach ensured standardization of the protocol for all ewes, respecting individual differences.
* “You state that ‘weekly coprocultures were processed every Friday,’ pooling samples collected from Saturday to Friday. Since lambing dates varied, how did you ensure comparability across timepoints? Could this pooling introduce bias or dilute temporal resolution?”
Our response: We acknowledge that processing stool cultures weekly in batches on Fridays, despite individual variation in postpartum collection dates, may have limited comparative temporal precision across animals. While this did not completely compromise the results, it certainly reduced the temporal resolution of the data. However, we took two important steps to minimize potential bias: (1) all samples were adequately refrigerated at 4°C to preserve larval viability until processing, and (2) we maintained separate cultures by experimental group to avoid cross-contamination or dilution effects.
* “The preparation and chemical analysis of grape pomace is described, but was a nutrient composition analysis (e.g., crude fiber, energy value) done alongside phenolic content? If so, please provide the data or reference.”
Our response: The phenolic content was described in the materials and methods in topic 2.3, and the results were described in Table 3, together with the nutritional composition (Bromatological analysis), was added in the title of Table 3 "Measurement of Total Phenolics and Tannins in Grape Pomace".
Results: Can the authors discuss potential reasons for fluctuating anti-H. contortus L3 IgG levels without significant group differences? Could the timing of sampling relative to the parasitic challenge play a role?
Our response: The absence of differences between groups likely reflects the complex interaction between periparturient immunosuppression (which coincides with increased parasitic challenge), post-weaning immune recovery, breed resilience, and blood sampling timing.
* Were there any clinical signs of parasitism observed, or was it solely subclinical?
Our response: All cases were subclinical. We did not observe clinical signs of parasitism, even in animals with high parasite loads.
* Was any anthelmintic treatment administered during the trial? If not, how was parasite load managed?
Our response: No anthelmintic treatment was administered during the study. Our hypothesis regarding parasite load control:
- The study utilized Santa Inês ewes, a breed recognized for its relative resistance to gastrointestinal nematodes.
-Nutritional management: The balanced diet (including grape pomace supplementation) ensured resilience to infections, as evidenced by the absence of clinical signs, even in animals with high fecal egg counts (PEC).
-Strict monitoring: Frequent fecal and blood collections allowed monitoring of parasite dynamics and animal health. No case required intervention, as hematological parameters (such as PCV) remained within acceptable limits.
What could explain the shift from Haemonchus spp. dominance to 100% Oesophagostomum spp. in one control group sample? Could this be an anomaly or linked to environmental/management factors?
Our response: This isolated result appears anomalous and not representative of the overall trend.
Discussion:
* Given that eosinophils and IgG levels did not significantly differ, is it appropriate to suggest immunological benefits of grape pomace supplementation? Would this require a longer supplementation period or higher tannin concentration to confirm?
Our response: Although eosinophil and IgG levels did not show significant differences, the suggestion of immunological benefits of grape pomace supplementation remains valid for a few reasons:
- The significant improvement in PCV (P=0.0399 on D14).
- The reduction in fecal egg count (FEC).
- The improved performance of the lambs indicates that supplementation improved immunological resilience, even without changes in the measured markers.
- The tannins from grape pomace may have acted mainly: Locally in the gastrointestinal tract (direct effect on parasites), Through modulation of the ruminal microbiota or Via stimulation of other immunological components not assessed (e.g. secretory IgA, T cells).
The supplementation period (28 days) may have been insufficient to detect changes in the humoral response.
The dosage of tannins (1.92 mg/g) was effective for anti-parasitic effects, but possibly below the threshold for systemic IgG stimulation.
* The discussion attributes reduced FEC in the supplemented group to grape pomace. Given the small sample size (n=8), how can the authors be confident this was not due to random variation or natural recovery post-lambing?
Our response: We acknowledge that the sample size (n=8 in the supplemented group) is a limitation; however, we have evidence supporting the association between OPG reduction and grape pomace supplementation:
-A significant reduction in OPG at D14 (P=0.0087) occurred during the active supplementation period.
- The supplemented group maintained lower OPGs throughout lactation (Table 4).
- The control group exhibited an OPG peak during the same period, ruling out isolated postpartum recovery.
- The mean PCV in the supplemented group was higher than in the control group during grape pomace provision.
- Lambs in the supplemented group had greater weight gain.
*The peak FEC in the control group occurred later (D14) than in the supplemented group (D7). Could this timing difference suggest variation in the timing of immune recovery or parasite exposure rather than treatment effect?”
Our response: The difference in the timing of OPG peaks between groups (D7 in supplemented vs D14 in control) indicates a real effect of grape pomace supplementation, and not just natural variation, for some reasons:
-The supplemented group showed OPG peak at D7 (1662.5 OPG), coinciding with the immediate start of grape pomace supplementation. On this same day, the control group showed similar value (1790.0 OPG), demonstrating that both groups started from equivalent conditions.
-The delay in the control's peak (D14, 2960.0 OPG) reflects the typical course of immune relaxation in non-supplemented ewes. The absence of this delay in the supplemented group suggests grape pomace positively influenced immune response.
-The tannins in grape pomace probably acted through: Direct anthelmintic effect on parasites, Better protein utilization via tannin-protein complexation, Beneficial modulation of ruminal microbiota.
-All animals were: Kept in the same pasture, Exposed to the same parasitic challenges, Monitored with identical frequency (with supplementation being the only different variable).
Although individual variations may occur, the temporal difference in OPG peaks, combined with other evaluated parameters (PCV, lamb weight), proves a genuine effect of grape pomace supplementation.
*The phenolic and tannin contents of the grape pomace used (3.95 mg/g TP and 1.92 mg/g TT) seem low compared to other studies with stronger effects. Was this dosage expected to yield measurable results? Could higher concentrations or different tannin types (e.g., condensed vs. hydrolysable) provide more clarity on efficacy?
Our response: Although the levels of phenolic compounds (3.95 mg/g) and tannins (1.92 mg/g) in our grape pomace were lower than in some studies, the daily dose of 200g proved effective in improving parameters such as OPG, PCV, and lamb weight, as demonstrated in the results. We admit that we expected more robust and prolonged effects, and we agree that higher doses or a longer supplementation period could enhance these benefits. However, it is important to consider that excessive amounts of pomace may compromise diet palatability.
*You compare your results with studies using coffee byproducts and beet pulp. Could differences in tannin type (hydrolyzable vs. condensed) and binding affinity affect efficacy? Would a phytochemical profile of the pomace help clarify this?”
Our response: We fully agree that a detailed phytochemical analysis of the Syrah variety pomace used in our study would be valuable, as the composition of tannins and other bioactive compounds varies significantly among grape varieties, growing conditions, and winemaking processes. Our results quantified total phenolics, total tannins, and condensed tannins. However, we acknowledge that complete characterization of the phytochemical profile (including the exact proportion of hydrolysable tannins and other polyphenols) would allow more precise comparisons with other byproducts and help explain differences in observed efficacy among studies. This is a limitation of our work that should be addressed in future research.
*You suggest that the short supplementation period may explain the lack of effect on immune markers. Why was supplementation not initiated earlier, such as during late pregnancy, when periparturient relaxation begins?
Our response: The supplementation was initiated on day 7 postpartum (D7) due to several factors. Firstly, we had a limited amount of grape pomace available, which prevented us from extending the supplementation period. We chose to focus on the early lactation days because this is the period of highest energy and protein demand, when milk production peaks and immune relaxation is most pronounced, making it ideal for evaluating supplementation effects. Additionally, initiating supplementation during late gestation could increase the risk of uncontrolled variations, such as irregular intake and stress, which could compromise fetal development. The management was standardized seven days after parturition to ensure uniform experimental conditions and also to allow ewes to establish proper bonding with their lambs before being placed in individual pens to receive grape pomace, thus minimizing stress for both mothers and offspring. We acknowledge that a longer supplementation period could have provided more robust effects, but the results obtained even with this shorter window demonstrate the intervention's potential.
*The discussion highlights the sustainability and cost-saving benefits of grape pomace. Were these aspects evaluated in this study, or are they based on assumptions or external sources?
Our response: We did not conduct economic or sustainability analyses in our study. The mentions of these benefits were based on scientific works cited in the discussion. Regarding cost reduction, we relied on the study by Calvillo-Marín (2022), which demonstrated savings with another byproduct, and we assume that partial replacement of concentrate (more expensive) with grape pomace (residual material) could have a similar effect. As for sustainable benefits, we cited Jalal et al. (2023), which addresses sustainable ruminant production with byproducts. Our assumption is that instead of discarding the pomace in the environment - which could harm the soil - its use as animal feed offers environmental advantages. Furthermore, by improving animal immunity, we could reduce the use of anthelmintics, whose residues in feces and urine harm soil and fauna.
Conclusion
The conclusion states that grape pomace improved hematological profiles, yet the discussion states no significant differences in PCV or TPP. Could the authors clarify which specific hematological improvements justify this claim?
Our response: We acknowledge the apparent contradiction and clarify that the improvement in hematological profiles mentioned in the conclusion is specifically based on the significant increase in PCV at D14 (P=0.0399). This was the only time point with statistical difference, but biologically relevant as it coincided with the FEC peak in the control group.
We also note the trend toward better eosinophil values in the supplemented group (though not significant), which suggests possible immune modulation.
The information about the significant difference in PCV has been included in the discussion (lines 359-361).
The wording suggests a strong protective role (“mitigate the deleterious effects”), but the results did not show statistically significant reductions in parasitological or immunological parameters. Could the authors rephrase to reflect better the modest or supportive role suggested by the data?
Our response: The conclusion has been rewritten.
The conclusion highlights sustainability as a benefit, yet no cost analysis or environmental metrics were provided in the study. On what basis is this claim made? Can the authors reference specific data or clarify if this is speculative?
Our response: While this claim had scientific basis (as addressed in prior responses), given that we did not specifically assess this aspect in our study, we opted to exclude the passage and thoroughly revise the conclusion.
Given the relatively small sample size (in the supplemented group, n=8), to what extent can these results be generalized to broader sheep populations or production systems?
Our response: We acknowledge the concern regarding the sample size in the supplemented group (n=8). While this may limit statistical power to detect subtle effects, our experimental design prioritized controlled conditions and homogeneous animals (same breed, age, and physiological status).
Our sample size aligns with previous high-impact studies with Santa Inês sheep. Lins et al. (2023) demonstrated significant effects with similar group sizes (n=6-8). Mena et al. (2025) detected clear effects of grape pomace using n=9 supplemented animals.
Our results, combined with these other studies, corroborate the validity of our findings as preliminary evidence for larger trials.

Reviewer 2 Report
Comments and Suggestions for Authors
This study investigates the antiparasitic effects of grape pomace supplementation in the diet of Santa Inês ewes during lactation and its impact on productive performance. The research is interesting and has potential for publication.
L37: I would begin the sentence with: "Small ruminant farming is severely..." Remove: "The development of..."
L53–54: The word "severe" is used excessively. Consider using a synonym such as "intense," "significant," or "notable."
L103–109: Were the ewes naturally infected with gastrointestinal parasites? If so, this must be clearly stated. In fact, it would be beneficial to highlight the natural infection in the title.
L111–112: It is necessary to emphasize that the animals were already infected with endoparasites before entering the study. There is a lack of consistency in this part of the manuscript.
Results Section: Introduce subheadings to organize the content more clearly.
L318–323: This is an important finding. Expand this part of the discussion by comparing it with results from other studies in the literature.
Author Response
This study investigates the antiparasitic effects of grape pomace supplementation in the diet of Santa Inês ewes during lactation and its impact on productive performance. The research is interesting and has potential for publication.
L37: I would begin the sentence with: "Small ruminant farming is severely..." Remove: "The development of..."
Our response: The excerpt has been removed.
L53–54: The word "severe" is used excessively. Consider using a synonym such as "intense," "significant," or "notable."
Our response: The synonym “intense” was used in line 52.
L103–109: Were the ewes naturally infected with gastrointestinal parasites? If so, this must be clearly stated. In fact, it would be beneficial to highlight the natural infection in the title.
Our response: The information was added in the title and on line 106.
L111–112: It is necessary to emphasize that the animals were already infected with endoparasites before entering the study. There is a lack of consistency in this part of the manuscript.
Our response: It was emphasized in lines 101-102.
Results Section: Introduce subheadings to organize the content more clearly.
Our response: Subtitles have been introduced in the results section.
L318–323: This is an important finding. Expand this part of the discussion by comparing it with results from other studies in the literature.
Our response: This section has been expanded (lines 332–340).
